# A Direct-Reading MEMS Conductivity Sensor with a Parallel-Symmetric Four-Electrode Configuration

**DOI:** 10.3390/mi13071153

**Published:** 2022-07-21

**Authors:** Zhiwei Liao, Junmin Jing, Rui Gao, Yuzhen Guo, Bin Yao, Huiyu Zhang, Zhou Zhao, Wenjun Zhang, Yonghua Wang, Zengxing Zhang, Chenyang Xue

**Affiliations:** 1State Key Laboratory of Dynamic Measurement Technology, North University of China, Taiyuan 030051, China; s2006174@st.nuc.edu.cn (Z.L.); junmin-jing@outlook.com (J.J.); 18406583750@163.com (R.G.); gyz20000113@163.com (Y.G.); yao_bin2021@163.com (B.Y.); zshy980828@163.com (H.Z.); zzzzhou95@163.com (Z.Z.); wangyonghua@nuc.edu.cn (Y.W.); xuechenyang@nuc.edu.cn (C.X.); 2School of Aerospace Engineering, Xiamen University, Xiamen 361102, China; 13485461170@163.com

**Keywords:** conductivity sensor, polarization effect, temperature compensation, high accuracy

## Abstract

This work proposes a design for a direct-reading conductivity sensor with a parallel symmetrical four-electrode structure, which integrates a silicon-based platinum thin-film strip electrode and a serpentine temperature compensation electrode. The optimal structural parameters of the electrode were determined by finite element simulations performed via COMSOL Multiphysics. Next, the designed conductivity sensor chip was fabricated using MEMS technology, and subsequently, the conductivity measurement circuit was designed to test the fabricated sensor’s performance. In laboratory tests, the optimal AC excitation frequency was observed to be 1.067 kHz, while the maximum measurement range was 0–107.41 mS/cm and the measurement precision in low concentration range (0–76.422 mS/cm) was ±0.1 mS/cm. Furthermore, the maximum measurement error of the sensor evaluated using the National Center of Ocean Standards and Metrology was ±0.073 mS/cm. The designed sensor possesses the characteristics of high accuracy, high range, and miniaturization, and enables real-time reading of conductivity value and temperature compensation, which is of great significance for the on-site observation of the physical parameters of marine environment.

## 1. Introduction

The oceans cover about 70.8% of the Earth’s surface, and there are various abundant resources in the vast ocean field. The measurement of ocean temperature and salinity is crucial in studying the marine environment. Through these measurements, the ocean circulation, marine ecological environment, marine biodiversity, and marine energy development can be monitored and studied [1,2]. In 1974, the “The Practical Salinity Scale” in UNESCO defined salinity to be calculated based on seawater’s conductivity, temperature, and pressure, where pressure has a relatively minor effect on the salinity [3]. Likewise, measuring salinity by conductivity, temperature, and pressure has many advantages, such as high accuracy, fast measurement speed, high reliability, and easy on-site measurement [4]. In this regard, there exist a variety of commercial CTD (Conductivity, Temperature, Depth) sensors used in high-precision marine development, such as the Seabird series, which occupies a leading position in the global market [5]. However, its large size, high cost, and energy consumption limit its promotion in the marine ranching and three-dimensional marine environment monitoring [6,7,8].

From 2005 to 2007, Broadbent, a scholar at University of South Florida in the United States, focused on using the liquid crystal polymer (LCP) with low hygroscopicity and permeability to manufacture a substrate and resistive temperature sensor (RTD) [9]. In that work, the electroplated nickel-gold-platinum alloy was used as a flat film four-electrode conductivity cell to develop a miniaturized CTD system. Relevant experiments concluded that the temperature and conductivity measurement accuracies were ±0.546 °C and ±0.882 mS/cm, respectively. In 2005, S. Bhansali et al. designed a MEMS-based CTD sensor [10], in which conductivity sensors generally use a parallel-plate capacitor structure, where most of the electric field is confined between the two parallel plates. However, some of the electric field is still distributed at the edges of the plates. To prevent the influence of external electric field on the measurement accuracy, installing guard rings at both ends of a board or adding electrodes and auxiliary circuits is adopted.

In 2011, X. Huang et al. from University of Southampton, UK, developed a measurement system based on the MEMS process for a seven-electrode conductivity sensor and a platinum temperature resistance sensor [11]. In that work, a 500 μm thick glass substrate was coated with 100 nm thick platinum layer serving as electrodes, wires, and pads. The platinum layer was then covered with a 25 μm epoxy laminate (SY320) insulation. The conductivity and temperature measurement accuracies were up to ±0.03 mS/cm and ±0.005 °C, respectively. Meanwhile, the temperature sensor drifted by 0.1 °C while the conductivity drifted by about 5.003 mS/cm, after five weeks. In 2013, Myounggon Kim et al. from the School of Mechatronics, Gwangju Institute of Science and Technology proposed an integrated microfluidic-based sensor module for real-time monitoring of reverse osmosis (RO) that measures temperature, conductivity, and salinity. The microfluidic device was constructed from a thin metal film and a microfluidic channel that was fabricated using the microelectromechanical system (MEMS) technology [12]. Recently, in 2020, Wu Chaonan et al. of Ningbo University also proposed a conductivity and temperature sensor fabricated using MEMS technology, where the sensor chip size was about 12 mm × 12 mm, and 34 chips could be fabricated simultaneously on a 4-inch silicon substrate [13]. The developed chip has high sensitivity, fast response time, and a good repeatability of temperature measurement. The experimental results revealed that the MEMS-based CT sensor has a temperature sensitivity of 0.0619 °C/Ω, a cell constant of 2.559 cm^−1^, and conductivity and temperature measurement accuracies of ±0.08 mS/cm and ±0.05 °C, respectively, providing valuable experimental data for ocean measurements.

This paper proposes a conductivity sensor that integrates temperature-compensated electrodes with a parallel four-electrode structure. The optimal parameters of the sensor structure were determined by finite element simulation, and the design of the sensor package structure was completed. Simultaneously, the hardware circuit and data acquisition algorithm were designed. The sensor’s range, precision, accuracy, and consistency were tested in the laboratory and third-party testing institutions. Furthermore, the sensor can read the conductivity through the master computer.

## 2. Working Principle of Sensor

Mainly, there are two types of conductivity sensors for marine environment monitoring applications: electrode conductivity sensors [14,15,16,17] and inductive conductivity sensors [18,19,20]. The inductive conductivity sensor is electrodeless, and the metal part of the sensor is not in direct contact with seawater. Moreover, the non-metallic shell is not easy to corrode, and the influence of the polarization effect is also avoided. Therefore, an inductive sensor is more suitable for the field measurements with high stability in harsh environments. However, it is susceptible to the proximity effect, and consequently, the sensor’s electric field is easily disturbed or distorted by the surrounding objects. The inductive conductivity sensor uses a toroidal transformer, which inherently results in a large size of the sensor, thereby limiting its integration with other miniaturized MEMS sensors.

Alternatively, electrode conductivity sensors measure the conductivity through a conductivity cell whose parameters are closely related to the position and shape of electrodes. In addition, the electrodes are divided into excitation electrodes and measurement electrodes. The excitation circuit provides a constant AC signal to the excitation electrodes, and a stable electric field is generated in the conductivity cell. Here, the measurement electrodes detect the potentials in different areas inside the conductivity cell and output them after the signal processing.

The conductivity *σ* of seawater is a physical quantity that describes its current transport capability, essentially reflecting the level of electrolyte concentration in seawater. Conductance *G* is inversely proportional to resistance *R*, and is given by the ratio of current *I* and voltage *V*:(1)$$G=\frac{1}{R}=\frac{I}{V}$$

The formula for calculating conductivity can then be expressed as:(2)σ=κ⋅G
where *κ* is the cell constant, which can be expressed as [13]:(3)$$\kappa =\frac{2}{\pi l}\mathrm{arccosh}\frac{d}{2a}$$

Among them, *l* is the length of electrode, *d* is the distance between two inner electrodes, and *a* is the width of voltage electrode. Meanwhile, the four electrodes are axially symmetrically distributed.

When an electrode conductivity sensor operates in seawater, electrochemical reactions occur at the contact surface between seawater and metal electrodes, which cause electron transfer between seawater and electrode. Notably, the polarization effect is defined as the change in ion concentration around the electrode due to the electric field generated by electrode. At the junction of electrode and seawater, the molecules and ions partially dissolved in seawater will be adsorbed on the surface of electrode in the form of chemical bonds, to form an inner layer. Here, the electric center trajectory of these ions is called the inner Helmholtz plane (IHP). Furthermore, the cations in seawater are attracted by the Coulomb force of the anions on electrodes, and the electric center trajectories of these cations are called the outer Helmholtz plane (OHP). In addition, some ions and molecules are not adsorbed, and instead, they are distributed in the diffusion layer due to the influence of electric field force and thermal motion, as demonstrated in Figure 1a.

The contact surface between electrode and seawater plays a capacitance role when there is no charge transfer at the electrode–seawater interface. This capacitance is called as the double-layer capacitance *C_dl_*, formed by the Helmholtz capacitance *C_H_* in series with the equivalent capacitance *C_G_* of diffusion layer. When a voltage is applied to the electrodes, charge transfer occurs between the seawater and electrodes, and such form of charge transfer is similar to a leakage current through a double-layer capacitor, which can be in fact seen as an impedance in parallel with the double-layer capacitor—the Faraday impedance [21]. Accordingly, the Faraday impedance can be expressed as a series connection of charge transfer impedance *R_ct_* and Warburg impedance *Z_W_* [22], where *Z_W_* is expressed as:(4)$$Z_{W}=\frac{\sqrt{2}A_{W}}{\sqrt{j\omega }}$$
where *j* is an imaginary number, ω is the angular frequency of excitation signal, and *A_W_* is the Warburg coefficient [23].

When an AC excitation is applied to the excitation electrode, the contact surface between electrode and seawater can be represented by the double-layer capacitance *C_dl_*, charge transfer impedance *R_ct_*, and Warburg impedance *Z_W_*. Combined with the stray capacitance *C_P_* and seawater equivalent resistance *R_W_*, the equivalent circuit of the four-electrode conductivity sensor can be represented as shown in Figure 1b. When an AC excitation is applied between the two current electrodes, there will be a voltage drop across the equivalent resistance R_W2_ of the solution between the voltage electrodes. This voltage drop can be measured by a voltage electrode with high input impedance. The current through the solution can be measured by the current electrode, so that the conductivity of the solution can be measured.

In addition, as the temperature of solution changes, the mobility of ions in the solution is also affected, thereby resulting in a temperature-dependent change in the conductivity. Therefore, to overcome the influence of temperature and make the conductivity of different solutions comparable at different temperatures, the conductivity should be temperature compensated. The temperature compensation formula is provided in Equation (5) as:(5)$$\sigma_{15}=\frac{\sigma_{t}}{1+\beta \left(t-15\right)}$$
where *σ_t_* is the conductivity of solution at temperature *t* °C, *σ_15_* is the conductivity of solution at 15 °C, and *β* is the temperature coefficient of solution conductivity.

## 3. Design and Fabrication

### 3.1. Structure and Package Design

In this work, a 525 μm thick 4-inch P-type <100> silicon wafer with high accuracy, good consistency, and low cost is used as the substrate. Essentially, electrode design is the most crucial part in the design of silicon-based thin film conductivity sensor. In designing a multi-electrode conductivity sensor, each pair of electrodes has strict spacing requirements. To reduce the parasitic capacitance, positions of current and voltage electrodes should be strictly symmetrical. As shown in Figure 2, the four-electrode design separates the voltage electrode from the current electrode, which can further weaken the electrode polarization phenomenon. The proposed sensor integrates four parallel strip electrodes, serving as current and voltage electrodes. Furthermore, a temperature electrode with a twisted alignment is also designed between the two voltage electrodes. In addition, platinum metal with good chemical stability and corrosion resistance is used as the material for the electrodes. Its resistance value varies linearly with temperature, and hence, it is used as a temperature compensating electrode. Moreover, silicon nitride film is used to insulate and protect the temperature compensating electrode. Silicon nitride film has good mechanical properties, high dielectric strength, chemical stability, and low film stress. Thus, it can be an excellent protective and insulating layer for the temperature electrode.

The dimensions of the electrodes were determined using COMSOL FEM simulations. The chip was surrounded by seawater, and Figure 3a shows the potential distribution of the chip, where a clear potential difference between the two voltage electrodes can be observed. Notably, when the current density of conductivity sensor is too high, the electrodes may get damaged. In addition, increasing the electric field strength can improve the measurement accuracy. Therefore, when designing the electrode, the length of electrode should be appropriately selected to reduce the current density. While keeping the spacing and width of all electrodes constant, the effect of electrode length on the current density and electric field strength is illustrated in Figure 3b. As the electrode length increases, the current density decreases, while the electric field strength increases. Nevertheless, the current density should not be too small, otherwise, it will affect the measurement accuracy in the low range. With these considerations, the electrode length is determined to be 14 mm for this work. Additionally, for fixed electrode length and electrode spacing, the variation of current density with the width of current electrode is shown in Figure 3c. It can be seen from the results that when the current electrode width is 100 μm, the current density is the smallest, and likewise, the width of the current electrode is designed to be 100 μm. Since no current flows through the voltage electrodes, they can be as narrow as the manufacturing process allows. Correspondingly, the width of voltage electrode is set to 10 μm. When the potential difference between the voltage electrodes of conductivity sensor is higher, the sensor’s sensitivity will be higher. Evidently, the potential difference between the voltage electrodes varies with the electrode spacing, while keeping the electrode length and width unchanged, as elaborated in Figure 3d. Likewise, when the current electrode spacing is constant, the potential difference increases as the voltage electrode spacing increases. For a voltage electrode spacing of 4 mm, the potential difference is observed to be the largest, therefore the selected voltage electrode spacing is 4 mm. On the other hand, for a fixed voltage electrode spacing, the potential difference increases with the decreasing current electrode spacing. Accordingly, the current electrode spacing is set to 4.5 mm following the selection of voltage electrode spacing, while considering the manufacturing capabilities.

As shown in Figure 4b, epoxy resin is used here to encapsulate the chip adhered to the PCB, which can effectively prevent the exposed pads on the chip from corroding. Furthermore, the size of acquisition circuit board is designed to be 57.5 mm × 18 mm × 1.5 mm, as shown in Figure 4a. The protective cover made of acrylic material can prevent the probe from crashing, and the segmented package design allows a quick replacement of the probe. Moreover, the acquisition circuit board is installed inside the tube shell, and the assembled conductivity sensor is shown in Figure 4c. This package design can transmit data in real-time through cables or realize self-capacitive data storage through TF cards.

### 3.2. Structure and Package Design

A single chip integrates four parallel electrodes and a serpentine-shaped platinum thin-film resistor with a size of 17 mm × 7.5 mm × 0.5 mm, as shown in Figure 5b, and 61 such chips can be simultaneously fabricated on a 4-inch silicon wafer substrate. The fabrication process of the silicon-based thin-film platinum conductivity sensor chip is explained in Figure 5a. Initially, a 4-inch P-type <100> single-side polished silicon wafer with a 1-micron silicon dioxide film deposited on the surface was cleaned, and a low-stress silicon nitride film was grown on it by chemical vapor deposition (PECVD). Next, the photoresist was spin-coated, and the conductivity electrodes and temperature compensation electrode patterns were photo-etched on the photoresist by ultraviolet lithography (UVL). Then, conductivity electrodes and temperature compensation electrodes with a thickness of 300 nm were fabricated by electron beam evaporation (EBE) followed by the lift-off process. To reduce the influence of temperature compensation electrode on the measurement of seawater conductivity and prevent the temperature compensation electrode from deforming with the temperature changes, a silicon nitride film was deposited again on the temperature compensation electrode for protection and insulation. Since chemical vapor deposition deposits the silicon nitride over the whole surface, it is also necessary to use UVL and reactive ion etching (RIE) to pattern the silicon nitride protective layer.

### 3.3. Measurement Hardware and Algorithm

The design scheme for the processing circuit of conductivity sensor is shown in Figure 6. According to the measurement principle of four-electrode conductivity sensor, the master control chip generates and supplies a triangular wave AC signal to current electrode of the conductivity sensor. Meanwhile, a differential operational amplifier measures the voltage difference between the voltage electrodes with high input impedance. Next, the differential amplified output voltage is input to the master control chip for the integral operation through ADC collection. In addition, the current flowing through the other current electrode is input to the operational amplifier after being collected by the ADC, and also input to the master control chip for integral operation. In the master chip, the ratio of current and voltage obtained from the integration process is output through the serial port, and the output value is the original signal of the conductivity sensor.

## 4. Experimental Method

### 4.1. Conductivity Sensor Calibration

A triangular wave is used as an excitation to overcome the influence of double-layer capacitance and polarization effect. As shown in Figure 1b, when the excitation frequency is low, the impedance of C_P_ is big, and most of the current goes through R_W_. However, because of the voltage drop across the double-layer capacitances (C_dl_), the measured impedance magnitude will be higher than R_w_. This effect diminishes at increasing frequencies. However, at higher frequencies, the impedance of C_P_ will decrease, so that a part of the injected current will go through C_P_. Therefore, the impedance magnitude will be lower than R_w_ because of stray capacitance. Therefore, selection of an appropriate excitation frequency is critical in improving the measurement accuracy. To determine the proper excitation frequency, potassium chloride solutions of 0.35 mol/L, 0.40 mol/L, and 0.6 mol/L were prepared, and these three solutions were placed in a constant temperature bath with a temperature fluctuation of ±0.005 °C. Next, conductivity tests were conducted on the three solutions, and the changes in the circuit output with the concentration under different excitation frequencies are shown in Figure 7. It can be seen from the test results that when the excitation frequency is higher than 1500 Hz or lower than 937 Hz, the output of the circuit does not change significantly with the concentration. When the frequency is 1067 Hz, the output of the circuit is the largest, and the change with the concentration is the most obvious. Therefore, 1067 Hz is selected as the excitation frequency.

#### 4.1.1. The Laboratory Calibration

The output of conductivity sensor is the ratio of current and voltage after the integral and mean value processing, thus the conductivity sensor needs to be calibrated. During the laboratory tests, the sensor was preliminarily calibrated to take into account the influence of experimental environment. Different standard conductivity solutions were prepared with dried potassium chloride and deionized water, and the sensor’s output in these solutions was tested at 25 °C. As shown in Figure 8, from 0 to 77.08 ms/cm, the output of conductivity sensor changes linearly with the standard conductivity, and R^2^ = 0.99968. According to the fitting curve, the relationship between the output *X_O_* of conductivity sensor and the standard conductivity σ is:(6)$$\sigma =8.184\cdot X_{O}-0.265$$

#### 4.1.2. Third Party Mechanism Calibration

To overcome the impact of experimental environment, a calibration at a third-party organization was carried out, at the National Center of Ocean Standards and Metrology in China. Considering the temperature-dependent characteristics of conductivity of standard seawater, the conductivity sensor was placed in a pool of standard seawater, and data are collected at 5 °C intervals between 0 and 35 °C. The corresponding relationship between the output of the conductivity sensor circuit and the temperature is shown in Figure 9a. According to the calibration protocol, the conductivity of air is considered to be 0 mS/cm. Therefore, the fitted curve between the output of the circuit and the standard conductivity is shown in Figure 9b. The fitted relationship between the conductivity y and the output x of circuit is described in Equation (7).
(7)$$y=a_{0}+a_{1}\cdot x+a_{2}\cdot x^{2}+a_{3}\cdot x^{3}+a_{4}\cdot x^{4}+a_{5}\cdot x^{5}$$*a*_0_ = −0.02189, *a*_1_ = 3.23449, *a*_2_ = 5.94996, *a*_3_ = −2.03875, *a*_4_ = 0.29531, *a*_5_ = −0.01522

### 4.2. Temperature Calibration

Fundamentally, the resistance of thin-film platinum resistors designed with serpentine structure can reach up to 13 kΩ. Moreover, the resistance of thin-film platinum resistors changes linearly with temperature. According to the eight-point calibration method, the sensor was placed in a constant temperature bath with a temperature fluctuation of ±0.005 °C. Next, eight temperature points were set between 5–45 °C, YOWEXA’s YET-710 temperature sensor was used as the standard instrument, and the relationship between the resistance of thin-film platinum and the actual temperature was measured, as shown in Figure 10. The linear fitting relationship between the resistance of thin-film platinum resistance and the actual temperature is given as:(8)T=0.02548⋅X−309.0508

The *R*^2^ of the fitted curve is 0.99999, demonstrating that the indication value of thin-film platinum resistance possesses an excellent linear relationship with the temperature.

## 5. Results and Discussion

### 5.1. Range and Precision of Sensor

The conductivity of potassium chloride solution with different concentrations was tested using the developed silicon-based thin film conductivity sensor. As evident from Figure 11a, the sensor has a maximum range of 107.41 mS/cm. Each concentration of KCl solution was tested 60 times, and the corresponding standard deviation of the measured conductivity value is shown in Figure 11b. The measurement precision between 0 and 76.422 mS/cm ranges from ±0.005 mS/cm at 5.939 mS/cm to ±0.165 mS/cm at 76.422 mS/cm. Furthermore, the measurement precision between 81.879 mS/cm~107.41 mS/cm ranges from ±0.229 mS/cm to ±0.401 mS/cm.

### 5.2. Performance Consistency

Conductivity tests were carried out on potassium chloride solutions with different concentrations using three distinct silicon-based thin-film conductivity sensors. As shown in Figure 12, when the concentration is 0.1 mol/L, the error in the test results of three sensors is ±0.086 mS/cm. When the concentration is 0.3 mol/L, the error between Sensor 1 and Sensor 2 is ±0.012 mS/cm, while the measurement error of Sensor 3 is ±1.697 mS/cm. The measurement error is because the temperature of thermostatic bath had not stabilized before the test was started. Therefore, it can be safely stated that the output of silicon-based thin-film conductivity sensor is consistent within the effective range.

### 5.3. Performance of Temperature Compensation and Salinity Testing

To reduce the effect of temperature on the conductivity test, the conductivity must be temperature compensated. The conductivity of standard seawater was tested from 0 °C to 40 °C. The conductivity values obtained at different temperatures were all converted to conductivity at 15 °C according to Equation (5). Figure 13 shows the actual conductivity of seawater at different temperatures and the conductivity after temperature compensation, with a maximum error of 0.1917 mS/cm after the compensation. Conductivity and temperature can be measured simultaneously by the sensor and displayed directly on the master computer. The temperature, conductivity, and salinity indications can be read out in real time through the master computer, and the sampling time and sampling frequency can also be set through the master computer. When the test was performed in the laboratory, the sensor was fixed about 10 cm below the water surface. Here, the pressure depends on the depth, hence the salinity of the solution, can be calculated directly in the master computer. The developed sensor essentially enables the integrated testing of conductivity, temperature, and salinity.

### 5.4. Sensor Accuracy Test and Performance Comparison

The accuracy of conductivity sensor was tested at the National Center of Ocean Standards and Metrology. In the experiment, the conductivity of air and the conductivity of standard seawater at 0 °C, 5 °C, 10 °C, 15 °C, 20 °C, 25 °C, 30 °C, 35 °C, and 40 °C were tested. In comparison with the standard conductivity, the maximum measurement error of our sensor was ±0.073 mS/cm at 57.6772 mS/cm, as illustrated in Figure 14.

The performance comparison of the proposed sensor and other conductivity sensors using MEMS technology for marine measurement is presented in Table 1. It can be seen that the accuracy of our sensor is better than that of Hyldgrad multi-sensor system and Chaonan Wu’s CT sensor. The measurement range of the proposed conductivity sensor is larger than the other sensors. The chip size is also smaller than Broadbent’s PCB MEMS CTD size.

**Table 1 micromachines-13-01153-t001:** Comparison of our conductivity sensor and other sensors.

Sensor	Accuracy	Range	Chip size
Hyldgrad multi-sensor system [23]	±0.6 mS/cm	-	4 mm × 4 mm
Broadbent PCB MEMS CTD [24]	-	2–70 mS/cm	18 mm × 28 mm
Huangxi CT sensor [11]	±0.03 mS/cm	25–55 mS/cm	10 mm × 20 mm
Chaonan Wu CT sensor [13]	±0.08 mS/cm	0–101 mS/cm	12 mm × 12 mm
Our conductivity sensor	±0.073 mS/cm (0–70 mS/cm)	0–107.41 mS/cm	17 mm × 7.5 mm

## 6. Conclusions

This paper proposes a direct-reading MEMS conductivity sensor with four parallel electrodes, integrating temperature-compensated electrodes for real-time temperature compensation. The sensor can directly read out the measured conductivity, temperature, and salinity through the master computer. The developed sensor exhibited good consistency with 61 chips successfully fabricated on a 4-inch silicon wafer. Furthermore, the measurement circuit and algorithm of conductivity sensor were also developed, and an integrated package of sensor probe and circuit was realized. The maximum measurement range of the sensor in the laboratory was 107.41 mS/cm. Most importantly, a third-party standardized calibration was also carried out, and the accuracy of the conductivity sensor was better than ±0.073 mS/cm over the range of 0–70 mS/cm. Future research will focus on improving the measurement accuracy at a high range and the long-term stability of the sensor.

## Figures and Tables

**Figure 1 micromachines-13-01153-f001:**
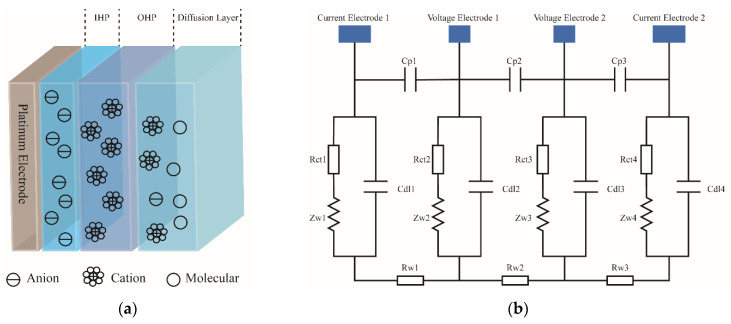
(**a**) A diagram demonstrating the structure of the electrode–electrolyte interface; (**b**) the equivalent circuit diagram of a four-electrode conductivity sensor with AC excitation applied to current electrode and voltage drop measured by voltage electrode.

**Figure 2 micromachines-13-01153-f002:**
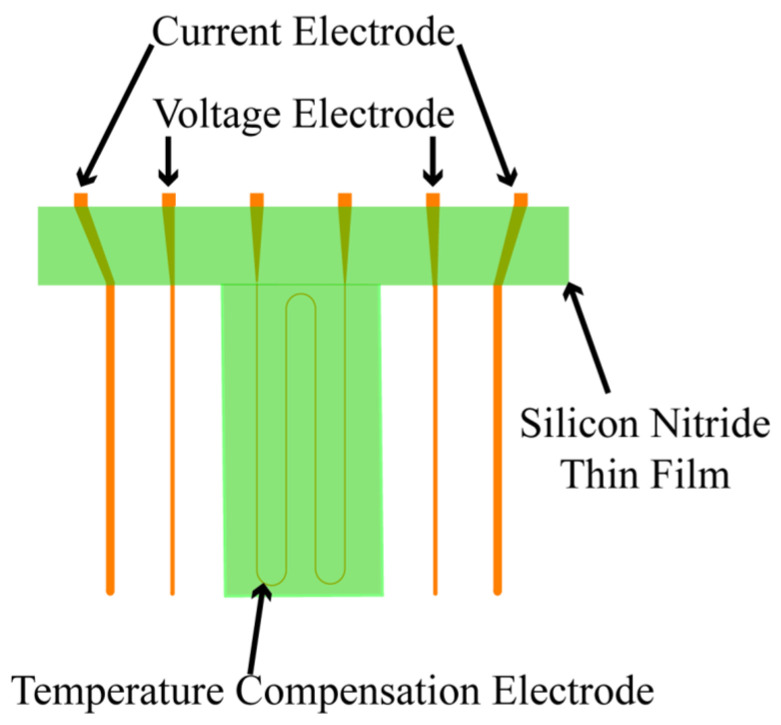
The figure is a schematic diagram of the chip structure.

**Figure 3 micromachines-13-01153-f003:**
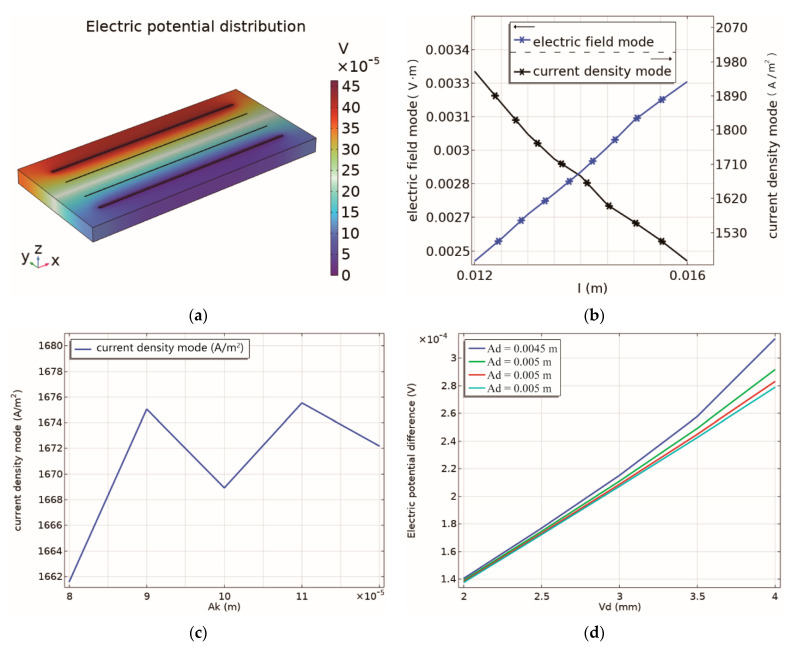
(**a**) The distribution of electric potential; (**b**) the effect of electrode length on current density and electric field strength; (**c**) the effect of current electrode width on current density; (**d**) the relationship between electrode spacing and voltage electrode potential.

**Figure 4 micromachines-13-01153-f004:**
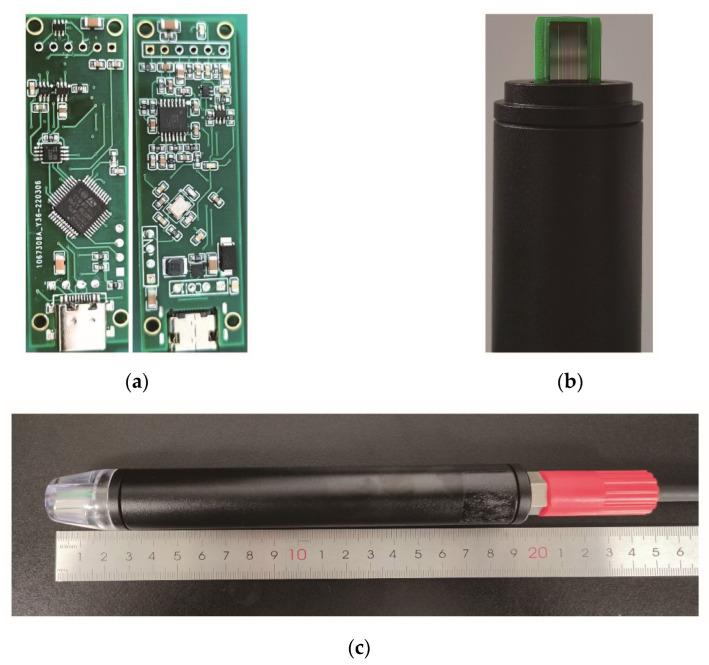
(**a**) The size of circuit board with double-sided patch is 57.5 mm × 18 mm × 1.5 mm; (**b**) a corrosion-resistant epoxy potted probe; (**c**) the assembled conductivity sensor with a length of 20 cm.

**Figure 5 micromachines-13-01153-f005:**
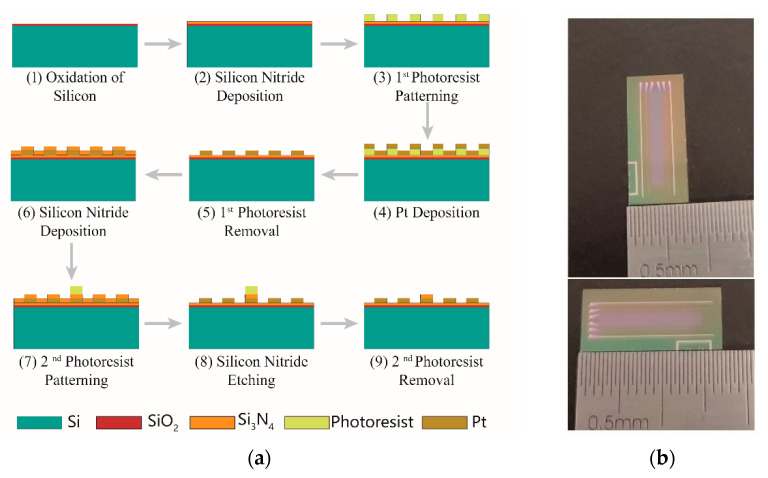
(**a**) The fabrication process of the conductivity sensor chip; (**b**) sixty-one chips, each with a size of 17 mm × 7.5 mm × 0.5 mm, can be fabricated on a 4-inch silicon wafer.

**Figure 6 micromachines-13-01153-f006:**
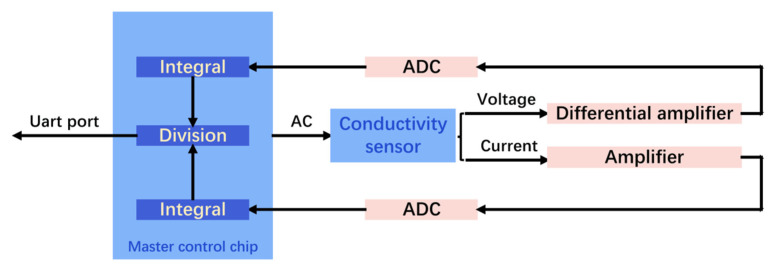
The designed circuit scheme of the four-electrode conductivity sensor. The signal is collected by ADC and processed by main control chip.

**Figure 7 micromachines-13-01153-f007:**
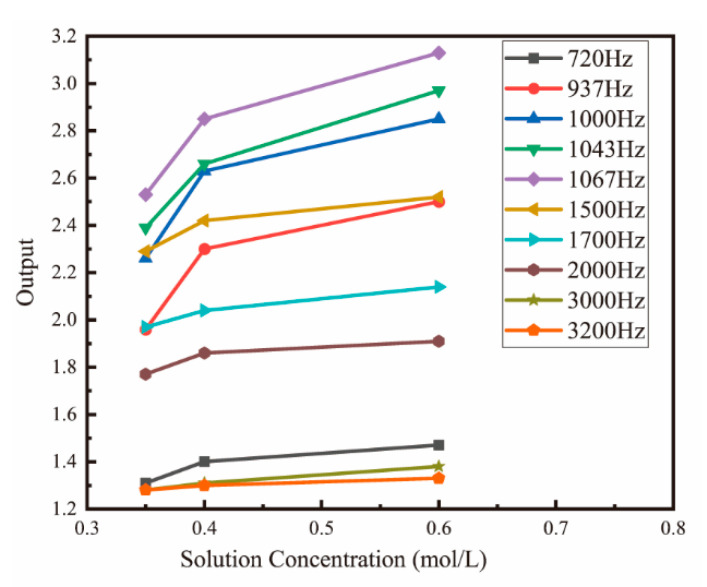
The relationship between concentration and the circuit output at different frequencies. From 720 Hz to 3200 Hz, the frequency is adjusted by dichotomy to determine the optimal excitation frequency.

**Figure 8 micromachines-13-01153-f008:**
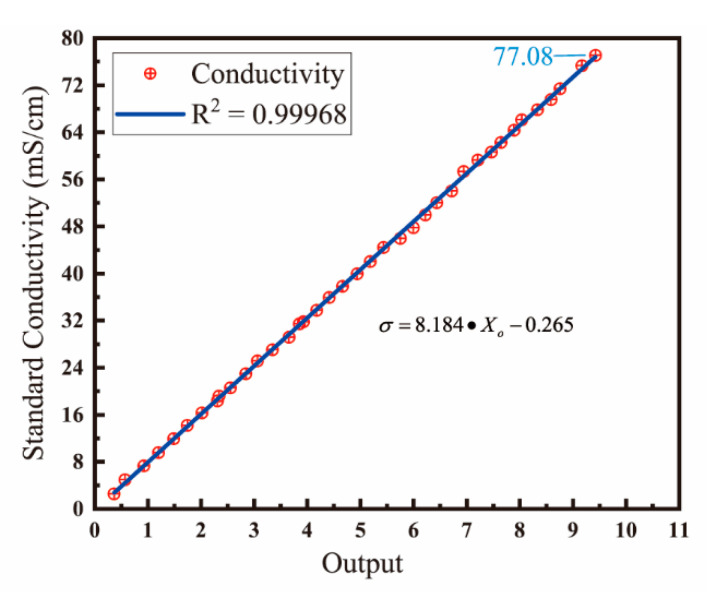
Fitted curve calibrated in the laboratory, calibrated by standard conductivity solutions with different concentrations.

**Figure 9 micromachines-13-01153-f009:**
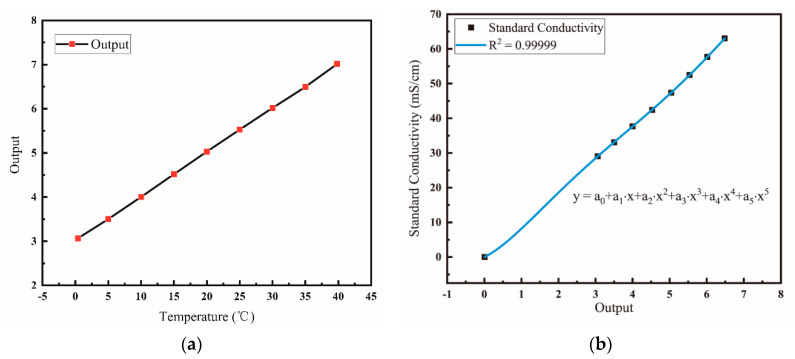
(**a**) Correspondence between temperature and circuit output. (**b**) Fitting curve calibrated by a third party, which is calibrated by changing the temperature of standard seawater.

**Figure 10 micromachines-13-01153-f010:**
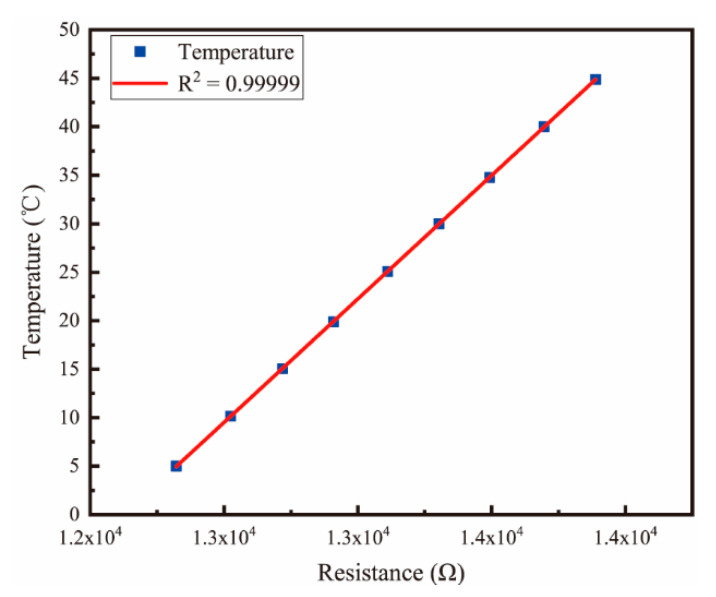
Variation in the resistance of platinum film resistor with temperature.

**Figure 11 micromachines-13-01153-f011:**
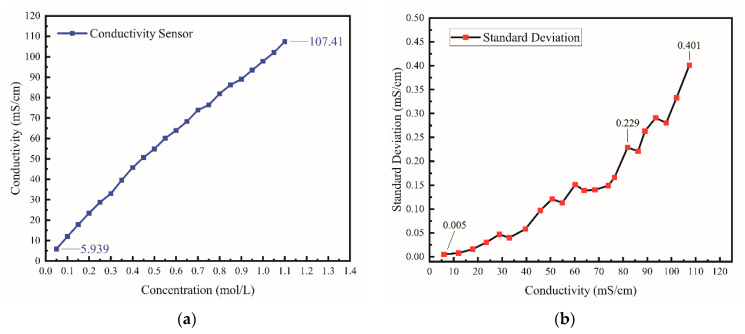
(**a**) The measurement range of our sensor. The maximum range of our sensor can reach 107.41 mS/cm. (**b**) The precision of our sensor, and the measurement precision in low and medium range is about ±0.1 mS/cm.

**Figure 12 micromachines-13-01153-f012:**
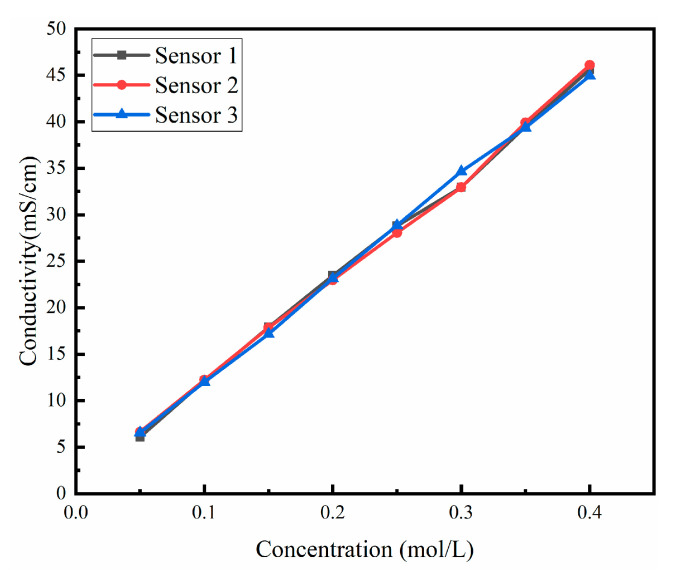
This figure shows the consistent performance of the proposed silicon-based four-electrode conductivity sensor based on platinum film. In different standard conductivity solutions, the test results of the three sensors are basically identical.

**Figure 13 micromachines-13-01153-f013:**
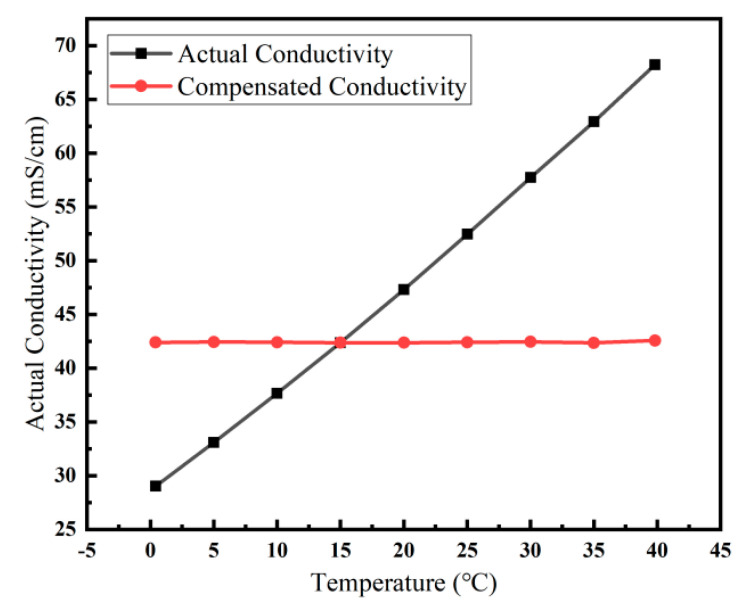
Measured values of the sensor at different temperatures compared to the temperature compensated values.

**Figure 14 micromachines-13-01153-f014:**
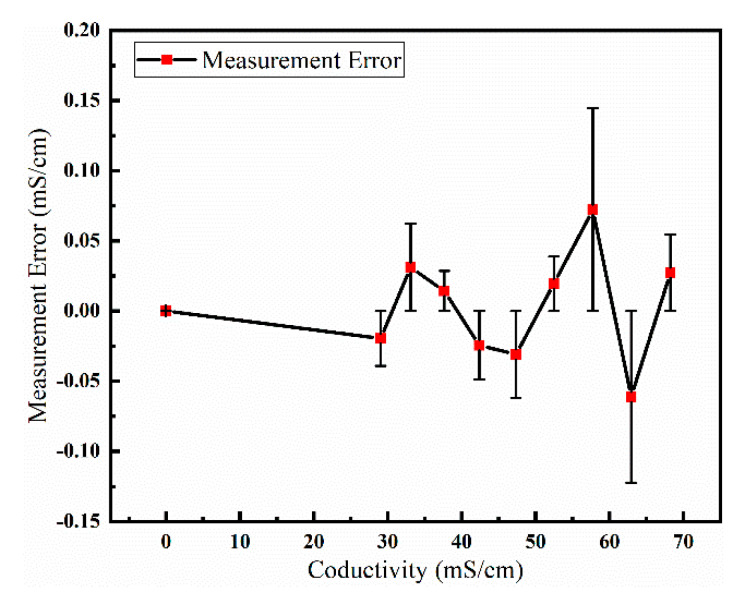
The graph shows the measurement error of the sensor.

## Data Availability

The data presented in this study are available on request from the corresponding author.

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
