# Peer review of "A Direct-Reading MEMS Conductivity Sensor with a Parallel-Symmetric Four-Electrode Configuration"

_micromachines, 2022, doi:10.3390/mi13071153_

Round 1

Reviewer 1 Report

The authors have demonstrated a design for direct-reading conductivity sensor with a parallel symmetrical Four-electrode structure. They also developed a completed system that can directly read out the measured conductivity, temperature, and salinity through the master computer. The overall form of the manuscript is good and concise. However, there are still some concerns and details that need to be clarified and modified before consideration for publication.

- The authors claim that their design possesses the characteristics of “high precision, high range and miniaturization, and enables real-time reading of conductivity value and temperature compensation.”. It will be more convincing if you can list the properties (e.g., precision, detectable range, and size) of the recently published works and compare them with your device performance.

- The authors did a great job in designing and optimising the electrodes specification, however, there was no explanation on the temperature compensation part. Please provide more information about that part.

- In section 4.1 “Conductivity Sensor Calibration”, the excitation frequency has an interesting effect on the circuit output. Is there any mechanism under this phenomenon? Is there any calculation to determine the optimum excitation frequency instead of manually tunning?

- There are grammatical errors that need to be fixed:

Please add space between the numerical value and its unit for example: “1.067 kHz” instead of “1.067kHz”.

Page 1, line 15, please un-capitalize “Four-electrode”.

Page 4, line 137 and line 150, “Where” should be “where” and non-indented.

Page 4, line 165, “for the electrodes.Its resistance value varies linearly with temperature, and hence, it is” missing space between sentences.

Page 7, line 224-225, “Since chemical vapor deposition deposits the silicon nitride over the whole surface, it is also necessary to use UVL and reactive ion etching (RIE) to pattern the silicon nitride protective layer once.”. Please remove “once”

In Fig. 5(a) step (h), should it be 2nd photoresist patterning?

Author Response

  • The reviewer’s comment:The authors claim that their design possesses the characteristics of “high precision, high range and miniaturization, and enables real-time reading of conductivity value and temperature compensation.”. It will be more convincing if you can list the properties (e.g., precision, detectable range, and size) of the recently published works and compare them with your device performance.

The authors’ Answer: Thanks very much for your comments, which are very helpful to improve the quality of this article. We added performance comparisons with other sensors in lines 374 to 380 on page 13 of the manuscript. As follows:

The performance comparison of the proposed sensor and other conductivity sensors using MEMS technology for marine measurement is presented in Table 1. It can be seen that the accuracy of our sensor is better than that of Hyldgrad multi-sensor system and Chaonan Wu’s CT sensor. The measurement range of the proposed conductivity sensor is larger than the other sensors. Chip size is also smaller than Broadbent's PCB MEMS CTD size.

Table 1. Comparison of our conductivity sensor and other sensors.

Sensor

Accuracy

Range

Chip size

Hyldgrad multi-sensor system[23]

±0.6 mS/cm

‒

4 mm × 4 mm

Broadbent PCB MEMS CTD[24]

‒

2~70 mS/cm

18 mm × 28 mm

Huangxi CT sensor[11]

±0.03 mS/cm

25~55 mS/cm

10 mm × 20 mm

Chaonan Wu CT sensor[13]

±0.08 mS/cm

0~101 mS/cm

12 mm × 12 mm

Our conductivicy sensor

±0.073 mS/cm

(0~70 mS/cm)

0~107.41 mS/cm

17 mm × 7.5 mm

  • The reviewer’s comment:The authors did a great job in designing and optimising the electrodes specification, however, there was no explanation on the temperature compensation part. Please provide more information about that part.

The authors’ Answer: Temperature compensation is explained in lines 150 to 157 on page 4 of the manuscript. Temperature compensation refers to converting conductivity at various temperatures to ones at 15°C. Varying temperatures result in conductivity changes through altering ionic activity. The conductivities of multiple solutions at different temperatures are comparable by normalization.

  • The reviewer’s comment:In section 4.1 “Conductivity Sensor Calibration”, the excitation frequency has an interesting effect on the circuit output. Is there any mechanism under this phenomenon? Is there any calculation to determine the optimum excitation frequency instead of manually tunning?

The authors’ Answer: We have revised in lines 259 to 265 on page 8 of the manuscript: “As shown in Figure 1(b), when the excitation frequency is low, the impedance of CP is big, and most of the current goes through RW. But because of the voltage drop across the double-layer capacitances (Cdl), the measured impedance magnitude will be higher than Rw. This effect diminishes at increasing frequencies. However, at higher frequencies, the impedance of CP will decrease, so that a part of the injected current will go through CP. Therefore the impedance magnitude will be lower than Rw because of stray capacitance.”. According to literature research, we found that the optimal excitation frequency is between 102 Hz and 106 Hz. Therefore, we verified through a large number of experiments, and finally determined that the optimal excitation frequency of the proposed sensor is 1.067 kHZ.

  • The reviewer’s comment:There are grammatical errors that need to be fixed:

Please add space between the numerical value and its unit for example: “1.067 kHz” instead of “1.067kHz”.

Page 1, line 15, please un-capitalize “Four-electrode”.

Page 4, line 137 and line 150, “Where” should be “where” and non-indented.

Page 4, line 165, “for the electrodes.Its resistance value varies linearly with temperature, and hence, it is” missing space between sentences.

Page 7, line 224-225, “Since chemical vapor deposition deposits the silicon nitride over the whole surface, it is also necessary to use UVL and reactive ion etching (RIE) to pattern the silicon nitride protective layer once.”. Please remove “once”

In Fig. 5(a) step (h), should it be 2nd photoresist patterning?

The authors’ Answer: We are very sorry for our incorrect writing. We have made revisions in the manuscript.

Reviewer 2 Report

Please find the comments attached.
